# Evaluation of Spleen Swabs for Sensitive and High-Throughput Detection of Classical Swine Fever Virus

**DOI:** 10.3390/pathogens14080767

**Published:** 2025-08-03

**Authors:** Orie Hochman, Kalhari Goonewardene, Chungwon J. Chung, Aruna Ambagala

**Affiliations:** 1National Centre for Foreign Animal Disease, Canadian Food Inspection Agency, Winnipeg, MB R3E 3M4, Canada; orie.hochman@inspection.gc.ca (O.H.); kalhari.goonewardene@inspection.gc.ca (K.G.); 2Department of Veterinary Pathology, Western College of Veterinary Medicine, University of Saskatchewan, Saskatoon, SK S7N 5B4, Canada; 3Foreign Animal Disease Diagnostic Laboratory, Animal and Plant Health Inspection Services, United States Department of Agriculture, Plum Island Animal Disease Center, Greenport, NY 11944, USA; chungwon.chung@usda.gov; 4Department of Medical Microbiology and Infectious Diseases, Max Rady College of Medicine, University of Manitoba, Winnipeg, MB R3E 0J9, Canada; 5Comparative Biology and Experimental Medicine, Faculty of Veterinary Medicine, University of Calgary, Calgary, AB T2N 1N4, Canada

**Keywords:** classical swine fever, spleen swabs, RRT-PCR, high-throughput, laboratory, detection

## Abstract

Despite intensive eradication efforts, classical swine fever (CSF) remains endemic across South America, Europe, Asia, and the Caribbean, highlighting the need for more effective surveillance and detection methods. Reverse-transcription real-time polymerase chain reaction (RRT-PCR) is the fastest, and most sensitive assay for detecting CSF virus (CSFV) genomic material. Previously, we demonstrated that spleen swabs outperformed spleen homogenates for the detection of ASFV genomic material by RRT-PCR. In this study, we compared CSFV genome detection in paired spleen homogenates and spleen swabs generated using 49 frozen and 33 fresh spleen samples collected from experimentally inoculated pigs with acute infection. The results show that the CSFV genome detection in spleen swabs is comparable to that in spleen homogenates. The study also demonstrated that the CSFV genomic material can be detected in spleen swabs during early CSFV infections, and the viruses can be successfully isolated from the swabs. The use of spleen swabs instead of spleen tissue homogenates for CSF detection will reduce labor, decrease costs associated with reporting, and increase the diagnostic throughput.

## 1. Introduction

Classical swine fever (CSF) is a highly contagious, fatal hemorrhagic disease of both domestic and wild pigs, which continues to cause severe economic losses within swine industries around the world [1]. The causative agent, CSF virus (CSFV), is a small, enveloped, single-stranded, positive-sensed RNA virus belonging to the genus Pestivirus [2]. Despite intense vaccination and other eradication efforts, CSFV remains endemic in some countries in Asia, Europe, the Caribbean, and South America [3]. Pigs infected with CSFV usually develop fever, lethargy, diarrhea, vomiting, neurological signs, hemorrhages, and purple skin color. Those clinical signs cannot differentiate CSFV-infected animals from those affected by other hemorrhagic diseases, such as African swine fever (ASF), porcine reproductive and respiratory syndrome, dermatitis–nephropathy syndrome, pseudorabies, bacterial septicemia, etc. Therefore, for the accurate identification of CSF, laboratory diagnosis is essential.

CSF presents in acute, chronic, and persistent forms [4]. Acute CSF, caused by highly virulent strains, progresses rapidly, often leading to death within two weeks, while less virulent strains can cause chronic or persistent infections. Reverse-transcription real-time polymerase chain reaction (RRT-PCR) offers the most sensitive and specific assay available for the detection of CSFV genomic material in clinical samples, and therefore, is widely used to diagnose acute CSF [5]. RRT-PCR is highly scalable and therefore suitable for front-line, high-throughput screening of clinical samples from CSF-suspected animals. The World Organization for Animal Health (WOAH) Manual of Diagnostic Tests and Vaccines for Terrestrial Animals recommends the use of RRT-PCR to detect CSFV genomic material in a variety of sample matrices, such as EDTA whole blood, serum, tonsils, and a range of tissues, including spleen samples [6]. Extraction of nucleic acid from tissue samples requires homogenization, which needs additional equipment, time, and labor. Recently, we demonstrated that spleen swabs are a better alternative to spleen homogenates for ASF virus genome detection [7]. The study demonstrated that spleen swabs were not only more sensitive, but also enhanced the ability to isolate ASFV from positive spleen samples. The co-detection of ASFV and CSFV is routinely performed and is critical for effective swine surveillance for countries free of these pathogens.

In this study, we compared traditional 10% spleen tissue homogenates with spleen swabs for nucleic acid extraction, CSFV genome detection by RRT-PCR, and virus isolation. Spleen tissues used in this study were collected from pigs experimentally infected with five different CSFV strains along with healthy known CSFV-negative domestic pigs. This study also aimed to determine the effect of using fresh vs frozen spleen tissue samples on CSFV detection in spleen swabs.

## 2. Materials and Methods

### 2.1. Samples

The study used spleen samples collected from pigs experimentally inoculated with multiple CSFV strains at the NCFAD Biosafety Level 3-Ag animal facility, Winnipeg, MB, Canada and Plum Island Animal Disease Center (PIADC), Orient, NY, United States. The virus strains include CSFV Koslov (Genotype 1.1, KF977610), Brazil (Genotype 1.5, PV453686), Pinillos (Genotype 2.6, OL963696) at the NCFAD, and Brescia (Genotype 1.1, AF091661) and Haiti (Genotype 1, unpublished) at the PIADC. The animal experiments at the NCFAD were conducted under the guidelines of the Canadian Council for Animal Care with approval from the Animal Care Committee at the Canadian Science Centre for Human and Animal Health (Animal user documents: AUD C-21-006, AUD C-22-001, AUD C-23-001). The animal experiments at the PIADC were conducted under the Institutional Animal Care and Use Committee (173.01-22-P). The samples were used immediately after collection or were frozen at −80 °C. Before use, the frozen spleen samples were thawed at room temperature. As known CSFV negative samples, a total of 20 domestic pig spleen samples collected under the ongoing CanSpotASF Canadian ASF surveillance program (https://www.animalhealthcanada.ca/canspotasf, accessed on 31 July 2025) were used.

### 2.2. Preparation of Tissue Homogenates and Swabs from Spleen Samples

At the NCFAD, spleen homogenates were prepared using a Precellys^®^ 24 Touch instrument (Bertin technologies, Rockville, MD, USA), as described previously [7]. Briefly, the spleen tissues were cut into small pieces (approximately 0.1 g) using sterile scissors or a scalpel. These cut tissue pieces were then weighed using a small laboratory scale, and 0.1 g of tissue was transferred into homogenization tubes, each containing 1 mL of sterile PBS. The tubes were then sealed and homogenized for 3 × 10 s at 5000 RPM, keeping the samples on ice between homogenization processes. The homogenates were then centrifuged at 4 °C, 2000× *g* for 20 min, and the resulting supernatant was then transferred to fresh sterile cryovials. Fifty-five µL of clarified supernatant from each homogenate was used for total nucleic acid extraction using MagMax™ Pathogen RNA/DNA kit (Cat# 4462359, Thermo Fisher Scientific, Waltham, MA, USA), according to the standard protocol provided by the manufacturer on the MagMax™ KingFisher Apex Deep Well Magnetic Particle Processor (Thermo Fisher Scientific, Waltham, MA, USA).

Spleen swabs were collected using sterile polyester-tipped applicator swabs (Cat# 25-806-1PD, Puritan Medical Products, Falmouth, ME, USA). Briefly, the spleen was cut open with a sterile scalpel and the cotton swab was inserted into the incision, pressed firmly, and twisted until the cotton tip was fully soaked. Cutting the spleen was not always necessary since some of the spleen tissues were fragile and the swabs could be inserted directly into the spleen by simply pressing the swab firmly against the tissue. The soaked ends of the swabs were then submerged into cryovials each containing 1 mL of sterile PBS. The excess swab length was snapped off, the cryovials were closed, vortexed for at least 10 s, and 200 µL of the liquid from each vial was used for nucleic acid extraction in accordance using MagMax™ Pathogen RNA/DNA kit (Thermo Fisher Scientific, Waltham, MA, USA) with the low cell-count-optimized protocol provided by the manufacturer of the MagMax™ KingFisher Apex Deep Well Magnetic Particle Processor (Thermo Fisher Scientific, Waltham, MA, USA). At the PIADC, the same swab collection and nucleic acid extraction procedures were used as at the NCFAD. Following the initial nucleic acid extractions, the remaining samples were stored at −70 °C for further testing.

Two different types of spleen swabs were collected and compared in this study. Spleen swabs taken during post-mortem from fresh spleen tissues, and those collected from previously frozen whole-spleen tissues.

### 2.3. RRT-PCR Detection of CSFV Genomic DNA and β-Actin Nucleic Acids

At the NCFAD, the detection of CSFV genomic material in both tissue homogenate and spleen swab samples was achieved using two different quantitative RRT-PCR assays, the CSFV-FLI [8] and the CSFV-PIADC [9] RRT-PCR. Both assays target the conserved sequences of the 5′ non-translated region (5′ NTR) of the CSFV genome. The CSFV-FLI was the validated RRT-PCR used at the NCFAD, and the CSFV-PIADC was developed, validated, and used at the PIADC.

Detection of β-Actin nucleic acids within the samples was undertaken by a secondary assay (Moniwa assay), which was included to determine efficient nucleic acid extraction and amplification [10]. All RRT-PCR assays were run as single plex assays, and the reactions were prepared using TaqMan™ Fast Virus 1-Step Master Mix (Cat# 4444434, Thermo Fisher Scientific, Waltham, MA, USA) and amplified using the Bio-Rad CFX96 instrument (Bio-Rad, Mississauga, ON, Canada), using the recommended cycling conditions (50 °C for 5 min; 95 °C for 20 s, followed by 95 °C for 3 s and 60 °C for 30 s) for the TaqMan™ Fast Virus 1-Step Master Mix for CSF FLI assay, and 50 °C for 30 min, followed by 95 °C for 15 min, and 45 cycles at 94 °C for 15 s and 56 °C for 60 s for the CSF-PIADC assay. To determine Ct values, a positive control well was included, while a regression analysis of the positive control alone was used to determine a threshold cutoff for all experimental wells. The results were designated positive for threshold cycle (Ct) values < 40 and negative for Ct values ≥ 40 or negative for samples in which the threshold was not attained before cycling was completed at 40 or 45 cycles depending on the assay.

### 2.4. Virus Titration

At the NCFAD, a selected number of homogenates (*n* = 6) and swab samples (*n* = 6 fresh and *n* = 6 post-thaw swabs) were tittered on 96-well cell culture plates using porcine kidney (PK-15) cells. Briefly, 10-fold dilutions of the samples were inoculated on to 60–70% confluent PK-15 cells in α-MEM (Cat# A1049001, Thermo Fisher Scientific, Waltham, MA, USA) supplemented with 1% Gentamicin (Cat#15750060), 1% Glutamax (Cat# 35050061), and 2% γ-irradiated horse serum (Cat# SH3743IH254, Fisher scientific, Ottawa, ON, Canada). Following 3 days of incubation at 37 °C and 5% CO_2_, the plates were fixed and stained with an anti-CSFV polyclonal antibody anti-CSFV polyclonal pig serum and HRP-conjugated goat anti-pig monoclonal antibody (Cat# 114-035-003; Jackson ImmunoResearch, West Grove, PA, USA).

### 2.5. Statistical Analysis

The Pearson correlation coefficient between the Ct values for the homogenate and the swab samples was calculated using the Graph Pad Prism Software, version 9.0.2 (San Diego, CA, USA).

## 3. Results and Discussions

Active integrated diagnostic surveillance for both ASFV and CSFV is an extremely important component in swine-producing countries that are free of both these high consequence pathogens. Recently, we reported [7] that spleen swabs are a better alternative than spleen homogenates for ASFV genome detection and virus isolation. The ability to utilize splenic swabs for the detection of both CSFV and ASFV in parallel would enable the animal health laboratories to reduce the time, cost, and the labor associated with sample processing and diagnostics.

In this study, we first evaluated 49 frozen spleen samples, collected from 25 pigs acutely infected with highly virulent CFSV Koslov [11], 20 pigs with moderately virulent CSFV Pinillos [11], and 4 pigs with low-virulent CSFV Brazil 2019-571 [Unpublished] (Table 1).

Along with the samples, 20 additional known CSFV-negative spleen samples that were collected under the ongoing CanSpot Canadian ASF surveillance system were processed (Table 2). The splenic tissue samples were thawed, a piece was cut and processed into a 10% (*w/v*) tissue homogenate, and a swab was collected from the remaining tissue. All samples were tested for the presence of CSFV genomic DNA using CSFV-FLI and CSFV-PIADC RRT-PCR assays and for porcine ß-Actin mRNA (endogenous internal control) by the Moniwa ß-Actin RRT-PCR assay.

ß-Actin, an endogenous internal control representing effective sample collection and extraction, was detected at comparable levels between the paired spleen homogenate and spleen swab samples (Table 1 and Table 2). The mean Ct values for ß-Actin detection in the experimentally infected tissue homogenate sampling set in Table 1 was 17.99 ± 0.58, whereas the swabs sampling sets mean Ct value was 17.18 ± 0.75. The known-negative spleen tissue samples showed a similar trend when compared between the two sampling types, where the tissue homogenate mean Ct value for ß-Actin detection was 21.95 ± 0.39 and the swab sampling sets mean Ct was 22.35 ± 0.55. These relatively comparable Ct means between the two methods of sample preparation suggests that the amount of tissue tested either as a homogenate or spleen swab were roughly equal. This demonstrates the effectiveness of spleen swabs in collecting viable diagnostic samples for nucleic acid extraction, compared to results obtained through tissue homogenization.

A similar trend was observed with the CSFV genomic detection by both CSFV RRT-PCR assays. Both FLI and PIADC assays were able to detect CSFV genomic material in all the sample types and across all genotypes. Tissue homogenate samples slightly outperformed spleen swabs across most samples but Ct values were near comparable levels. The CSFV-FLI RRT-PCR assay had a mean Ct-value of 21.41 ± 1.69 for tissue homogenates compared to a mean Ct-value of 21.88 ± 1.72 for spleen swabs. For the CSFV-PIADC RRT-PCR assay, mean Ct-value for homogenates was 19.23 ± 1.77 and for the swabs was 20.06 ± 2.33. The slight difference in mean Ct-values between the FLI and PIADC assays could be related to the inherent sensitivity between these two assays towards the CSFV genotypes tested. Despite the slight difference in Ct values, both RRT-PCR assays exhibited the same pattern of closely related detection levels of CSFV genomic material in the paired tissue homogenate and spleen swab. This can be seen in the strong positive correlation observed between the homogenates and the swabs (Figure 1).

The results above show that spleens collected from an acutely infected animal and frozen can be effectively swabbed and sufficiently yield adequate levels of both samples (indicated by β-Actin detection) along with target virus detection (CSFV in this case). Next, we compared CSFV and β-Actin detection in spleen swabs vs 10% spleen tissue homogenates prepared from fresh spleen tissues collected at the post-mortem examinations from pigs acutely infected with CSFV. The success of this collection strategy would allow disease investigators to directly swab suspected animals and transport swabs instead of tissue samples. In this study, PIADC used 6 and NCFAD used 33 fresh spleen samples.

At the PIADC, from each spleen, three individual swabs and three individual tissue samples were collected and tested from acutely infected swine (Table 3). The three swabs and three tissue samples were collected representing the central, the upper, and the lower quadrants of each spleen. For the detection of CSFV genomic material, the PIADC RRT-PCR assay was used. The Vetmax Xeno Internal Positive Control (IPC) DNA was used as the exogenous internal control to monitor successful nucleic extraction and presence/absence of PCR inhibitors in the extracted nucleic acid samples.

In line with the findings from frozen spleen samples, the detection of CSFV genomic material in freshly collected spleen swabs and homogenates was comparable (Table 3). The mean Ct-value for the 18 homogenates was 20.21 ± 0.60, whereas the mean Ct-value for the 18 spleen swabs was 21.89 ± 0.59. The Ct values for the triplicate swab samples were also comparable, indicating that the swabs can be collected from any part of the spleen without affecting the results. The comparable Ct values among triplicate samples also confirm that the swabbing method is consistent. The Ct values for the exogenous internal control were also comparable between swabs and tissue homogenates, indicating that there were no PCR inhibitors in the samples.

The fresh swabs and tissue homogenates collected from 33 fresh spleen samples at the NCFAD were tested for CSFV genomic material by the FLI-CSFV RRT-PCR assay (Table 4). As seen with frozen spleen samples, the CT values for swab samples were comparable to the corresponding homogenates. The mean Ct-value for the 33 tissue homogenates was 25.93 ± 1.88 compared to 25.73 ± 2.26 for the swab samples. The mean Ct-value for beta-actin was 18.18 ± 0.66 for the homogenates and 17.02 ± 0.88 for the swabs.

The Ct values for CSFV genomic material in swab samples and homogenates from both laboratories showed a strong positive correlation similarly to that observed for the frozen samples (Figure 2).

The slight difference observed in CSFV genome detection between paired swab samples and tissue homogenates could interfere with early detection. Therefore, in the next experiment, we evaluated spleen swabs and homogenates collected between 2 and 8 days post-CSFV infection (Table 5). The spleen samples were collected from an ongoing experiment in which two groups of pigs were simultaneously inoculated with either CSFV Pinillos or CSFV Brazil 2019-571. Three to four pigs from each group were euthanized on days 2, 4, 6, and 8 post-infection and spleen samples were collected. Fresh spleen swabs were collected from each spleen harvested and the remaining splenic tissue was frozen at −80 °C. After 2–3 weeks, the spleen tissues were thawed, and post-thaw spleen swabs were collected. All samples were tested for the presence of CSFV genomic DNA using the CSF-FLI assay and ß-Actin using the Moniwa RRT-PCR assay.

The CSF FLI RT-PCR assay was able to detect CSFV genomic material in 3 out of 3 spleen homogenates vs only 2/3 spleen swabs from pigs infected with CSFV Pinillos on 2 dpi. The homogenate from pig #3 was very weakly positive (Ct 37.77) and the paired fresh and post-thaw swabs tested negative. In the CSFV Brazil 2019-571-infected pigs, the assay was able to weakly detect (Ct 38.11) CSFV genomic material in 1/3 spleen homogenates on 2 dpi but not in fresh or post-thaw spleen swabs. Starting from 4 dpi, CSFV genomic material was detected in all spleen sample types. The slightly increased detection in homogenates vs spleen swabs could be due to the more efficient release of CSFV virions/genomic RNA during the homogenization process, especially when extremely low levels of the virus are present in the splenic tissues. The comparable ß-Actin levels in all samples across all sampling types showed consistency in splenic swab sampling. The detection levels of CSFV genomic material and ß-Actin were comparable between fresh and post-thaw spleen swabs. This provides the diagnostic laboratories the option to freeze samples for later testing them using swabs.

Overall, in this study, CSFV genomic material was detected in 120 spleen swabs versus 122 paired spleen homogenates collected from acutely infected pigs, resulting in 98.36% sensitivity for spleen swabs compared to the spleen homogenates. The use of spleen swabs did not result in any false positives, and therefore, the specificity of CSFV genome detection in spleen swabs remained 100%.

Virus isolation is critical and remains the gold standard for the identification of viral pathogens, including CSFV. The ability to isolate virus from RRT-PCR-positive samples confirms that the samples contained viable virus, the isolated virus can be used for subsequent neutralization assays, and in pathogenesis experiments, to learn more about the phenotype of the virus. Virus isolation is a laborious process, and therefore, we selected a few samples that showed the lowest Ct values (samples collected on 2 dpi from the above-described CSFV Pinillos and CSFV Brazil 2019-571 experiments). As seen in Table 6, in agreement with the RRT-PCR Ct values, we were able to isolate virus from the homogenate and swab samples. When the splenic tissue contains very low CSFV, the spleen homogenate may test weakly positive/suspicious (Ct value above 36), and the paired swab sample negative, and in that case, isolating virus from the swab sample may not be possible.

## 4. Conclusions

Effective strategies in the detection of high-consequence viral pathogens, such as CSFV and ASFV, is critical in preventing catastrophic animal and economic losses. Two key areas of this are the development of a surveillance system and an outbreak response plan to ramp up high-throughput sample testing for the quick containment of viral spread. In this study, along with Cafariello et al. 2024 [7], we showed that fresh or post-thaw splenic swabs collected from acutely infected pigs are both capable and easily implementable sample type for high-throughput screening for CSF and ASF. In line with our findings, in a study conducted at slaughterhouses in Uganda, a high correlation (r = 0.8728) was observed between spleen tissue homogenates and spleen swab samples for ASFV genome detection by RT-PCR [12]. Many countries have already implemented passive surveillance programs for ASF and CSF, but current laboratory protocols call for spleen tissue samples to be processed into 10% homogenates before nucleic acid extraction, which is time-consuming and requires additional resources, limiting the number of samples the laboratories can realistically handle, thus significantly reducing throughput. The ability to use spleen swabs allows countries to easily collect large numbers of samples with minimal training and labor for sample processing without over burdening laboratories and outstretching their capacity to process these samples. Spleen swabs for ASFV detection have not yet been approved by the WOAH; however, the USDA has approved the use of spleen swabs as a sample type for official ASF testing in the USA, particularly during foreign animal disease (FAD) investigations [13]. In a small study conducted by Errington et al., swabs collected from several tissues, including spleen, thymus, lung, and bronchial lymph node samples, were tested for the detection of bovine viral diarrhea virus (BVDV) and porcine reproductive and respiratory syndrome virus (PRRSV) [14]. The results demonstrated that the swabs were a useful alternative to the traditional tissue lysis approach.

A large, scaled-up implementation of swab collection in the field was conducted during an Equine Influenza outbreak in New South Wales, Australia, by directly swabbing freshly cut tissue [15]. This resolved complications and delays associated with whole-tissue sampling, shipping, and processing, and the number of samples that were tested by animal health labs during the outbreak were dramatically increased. Similarly, spleen swabs can be collected and shipped to the laboratories during routine or outbreak surveillance for CSF and ASF. Swabs collected from spleen tissues can be immersed in viral transport media (VTM) and transported to the laboratories under refrigeration. When virus isolation is not required, the collected swabs can be immersed in molecular transport media, such as PrimeStore^®^ MTM (Longhorn Vaccines and Diagnostics, Bethesda, MD, USA) [16,17,18], which will inactivate the pathogens in the sample and stabilize the genomic material. This enables the samples to be stored and shipped at ambient temperature, eliminating the need for refrigeration and the risks of transportation of live CSFV and ASFV. Despite many advantages, the use of swabs in the field also faces some challenges, as it requires appropriate swabs, transport media, and ready availability of leak-proof tubes that can withstand the transport. Improperly closing or any damage to the tubes during transport could lead to complete loss of the samples and or cross-contamination.

## Figures and Tables

**Figure 1 pathogens-14-00767-f001:**
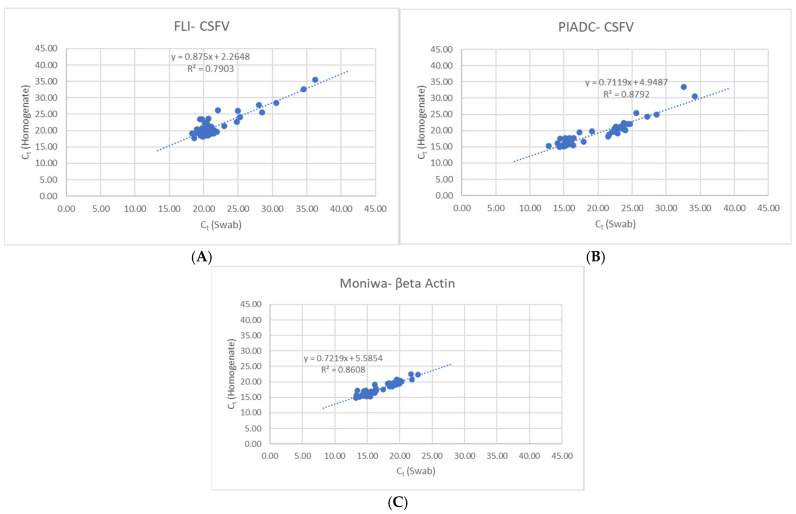
Strong positive correlation observed between 10% homogenates and spleen swabs by (**A**) FLI-CSFV, (**B**) PIADC-CSFV, and (**C**) Moniwa β-Actin assays.

**Figure 2 pathogens-14-00767-f002:**
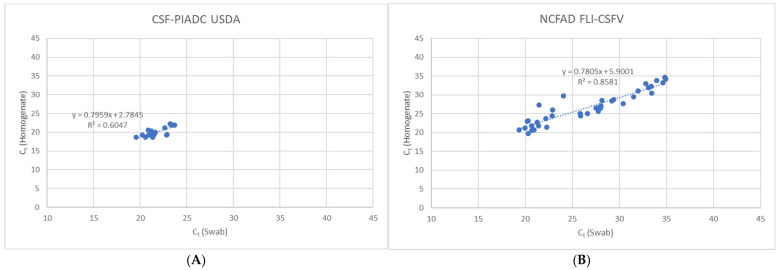
Strong positive correlation observed between freshly collected 10% homogenates and spleen swabs by (**A**) FLI-CSFV and (**B**) PIADC-CSFV.

**Table 1 pathogens-14-00767-t001:** RRT-PCR results for archived 49 frozen spleen samples collected from CSFV experimentally infected pigs at the NFCAD.

CSFV Strain	Pig	CSFV-FLI		CSFV-PIADC		Moniwa ß -Actin	
		Homo	Swab	Homo	Swab	Homo	Swab
CSFV Pinillos (Genotype 2.6)	1	19.65	21.88	20.69	23.81	18.94	19.32
	2	19.04	20.61	20.55	22.93	19.17	19.35
	3	18.98	20.66	20.64	22.70	20.78	19.54
	4	19.36	20.74	19.56	22.27	19.45	18.15
	5	19.47	21.23	20.78	23.67	18.46	18.44
	6	24.15	25.30	24.27	27.26	19.49	18.52
	7	19.48	21.42	21.16	23.71	18.82	18.69
	8	20.13	21.49	22.07	24.40	19.45	19.29
	9	18.42	20.60	20.28	22.55	18.45	18.81
	10	21.30	22.99	20.45	23.70	19.79	19.13
	11	20.86	20.36	21.17	22.62	20.73	19.59
	12	18.97	20.49	18.86	21.63	18.73	18.51
	13	18.94	20.68	19.19	22.83	19.24	19.25
	14	19.12	21.43	20.12	24.05	19.44	19.83
	15	20.59	20.81	19.69	22.09	20.43	20.02
	16	19.15	21.44	21.18	23.35	20.05	20.29
	17	19.97	21.48	21.99	24.69	19.38	19.41
	18	18.45	19.55	18.19	21.48	19.28	19.83
	19	19.15	20.65	20.57	22.44	19.62	18.40
	20	19.04	21.02	20.75	22.89	19.46	20.00
CSFV Koslov (Genotype 1.1)	1	19.31	18.97	15.21	15.10	17.16	16.40
	2	18.40	20.04	15.30	15.17	15.30	15.49
	3	19.19	18.33	15.24	12.76	16.48	15.60
	4	23.42	19.75	17.64	15.23	18.07	16.27
	5	20.15	19.46	16.09	15.07	14.78	13.21
	6	18.73	20.16	15.73	15.72	16.45	14.54
	7	18.20	19.88	15.29	14.66	15.52	13.98
	8	21.12	21.16	15.52	16.34	17.30	16.21
	9	27.87	28.01	22.32	23.75	17.20	13.45
	10	20.75	19.73	15.29	15.19	17.17	14.80
	11	17.76	18.59	15.09	14.98	16.16	15.34
	12	23.70	20.67	17.70	15.84	15.34	14.89
	13	21.49	20.29	17.73	16.45	15.89	14.36
	14	22.27	20.13	17.08	16.16	17.35	16.36
	15	26.16	22.05	19.38	17.27	19.09	16.14
	16	26.08	24.97	19.77	19.15	16.37	16.16
	17	19.41	19.80	14.96	14.35	16.10	14.33
	18	20.31	20.93	16.01	16.02	15.30	15.44
	19	22.22	20.43	17.09	15.32	15.65	14.73
	20	21.22	21.07	16.29	15.58	16.89	15.54
	21	19.19	19.63	15.38	15.21	15.70	13.30
	22	19.40	20.38	15.56	15.36	15.43	14.45
	23	20.46	19.00	16.15	14.05	16.38	14.46
	24	23.41	19.45	17.57	14.46	17.07	14.49
	25	22.67	24.80	16.58	17.89	17.54	17.42
CSFV Brazil 2019-571 (Genotype 1.5)	1	28.42	30.53	25.38	25.62	15.07	13.80
	2	35.50	36.25	33.44	32.55	20.67	21.89
	3	32.62	34.52	30.49	34.16	22.39	22.84
	4	25.62	28.53	24.88	28.61	22.54	21.73

**Table 2 pathogens-14-00767-t002:** RRT-PCR results for 20 frozen known negative spleen samples collected from CanSpot Canadian ASF surveillance system. ND = No detection.

	Pig	CSFV-FLI		CSFV-PIADC		Moniwa B-Actin	
		Homo	Swab	Homo	Swab	Homo	Swab
Known-Negative Spleens	1	ND	ND	ND	ND	22.19	22.60
	2					22.73	23.88
	3					21.12	22.14
	4					21.27	21.04
	5					21.74	22.56
	6					21.29	22.24
	7					20.58	19.42
	8					21.49	22.13
	9					20.83	20.12
	10					20.44	20.86
	11					23.06	23.77
	12					23.36	23.58
	13					21.76	22.26
	14					23.31	23.50
	15					22.33	23.34
	16					22.37	22.66
	17					22.49	22.26
	18					22.36	23.39
	19					22.29	22.31
	20					21.89	22.96

**Table 3 pathogens-14-00767-t003:** RRT-PCR results for 18 fresh samples collected from 3 experimentally infected pigs at the USDA-PIADC Facility. From each spleen, swab and tissue samples were collected in triplicate (depicted as 1–3) and tested individually.

CSFV Strain	Pig	CSFV-PIADC		Vetmax Xeno Internal Positive Control DNA	
		Homo	Swab	Homo	Swab
CSFV Brescia (Genotype 1.1)	10-1	21.16	22.61	33.96	32.84
	10-2	21.96	23.28	33.92	33.75
	10-3	21.92	23.70	34.05	34.06
	11-1	19.36	21.48	34.59	33.16
	11-2	19.29	20.88	34.26	33.31
	11-3	19.23	21.13	34.76	32.98
	12-1	19.39	22.85	33.99	35.91
	12-2	18.71	21.34	33.00	33.94
	12-3	19.25	22.81	34.08	33.78
CSFV Haiti (Genotype 1)	7-1	20.32	21.18	33.44	33.57
	7-2	20.64	20.84	33.55	33.21
	7-3	19.99	21.63	32.98	34.26
	8-1	22.26	23.19	34.76	33.51
	8-2	21.88	23.40	33.81	34.09
	8-3	21.86	23.45	33.98	33.78
	9-1	18.66	20.55	33.11	34.03
	9-2	18.65	19.56	33.18	34.42
	9-3	19.22	20.20	33.59	33.88

**Table 4 pathogens-14-00767-t004:** RRT-PCR results for 33 fresh spleen samples collected from CSFV experimentally infected pigs at the NCFAD.

CSFV Strain	Pig	CSFV-FLI		Moniwa β-Actin	
		Homo	Swab	Homo	Swab
CSFV Pinillos (Genotype 2.6)	5	28.51	28.12	18.9	17.14
	11	27.11	28.03	16.44	15.13
	16	29.70	24.05	19.53	16.68
	15	26.04	22.89	19.55	18.16
	22	25.59	27.76	18.54	18.32
	24	27.36	21.43	18.68	20.04
	4	22.73	21.28	18.67	18.69
	10	24.37	22.82	19.55	18.71
	20	23.04	20.18	19.8	19.27
	17	21.80	20.69	18.79	19.97
	18	23.13	20.30	19.42	18.46
	19	23.65	22.15	19.4	17.64
	7	20.65	20.93	19.36	18.72
	9	21.14	19.95	19.81	18.89
	12	20.71	19.32	18.73	19.35
	2	19.79	20.30	18.06	19.02
	6	21.72	21.38	19.59	17.54
	8	20.62	20.69	17.36	18.15
CSFV Brazil 2019-571 (Genotype 1.5)	4	29.46	31.48	15.05	14.82
	18	30.49	33.40	18.60	15.65
	23	32.26	33.38	15.01	13.47
	1	31.1	32.00	18.96	15.59
	3	26.59	27.73	17.34	15.21
	5	26.49	28.00	18.45	15.05
	6	25.08	26.61	19.03	19.27
	7	28.37	29.19	17.47	14.05
	11	25.04	25.84	18.45	15.26
	2	34.71	34.81	17.14	14.75
	8	27.71	30.39	18.62	15.02
	21	31.83	33.06	17.68	15.79
	9	33.02	32.77	14.57	14.97
	10	24.46	25.88	15.76	15.26
	17	21.4	22.26	17.79	17.54

**Table 5 pathogens-14-00767-t005:** Use of swabs vs homogenates for early detection of CSFV in spleen samples. The samples were collected from pigs simultaneously inoculated with CSFV Pinillos or CSFV Brazil 2019-571. ND = No detection.

Strain and Genotype	Pig Number	DPI	CSFV-FLI			Moniwa B-Actin		
			Homo	Fresh Swab	Post-thaw Swab	Homo	Fresh Swab	Post-thaw Swab
CSFV Pinillos (Genotype 2.6)	1	−3	ND	ND	ND	17.83	15.36	16.88
	13		ND	ND	ND	16.24	15.44	16.43
	14		ND	ND	ND	19.45	17.4	18.05
	3	2	37.77	ND	ND	19.82	18.08	17.45
	21		33.16	34.60	33.80	17.31	18.09	16.87
	23		34.15	34.92	36.23	18.53	15.24	16.6
	5	4	28.51	28.12	27.17	18.90	17.14	19.11
	11		27.11	28.03	27.17	16.44	15.13	16.63
	16		29.70	24.05	27.82	19.53	16.68	17.4
	15	6	26.04	22.89	24.40	19.55	18.16	18.82
	22		25.59	27.76	23.63	18.54	18.32	17.77
	24		27.36	21.43	31.29	18.68	20.04	20.28
	4	8	22.73	21.28	22.29	18.67	18.69	19.21
	10		24.37	22.82	22.84	19.55	18.71	18.65
	20		23.04	20.18	22.24	19.8	19.27	18.28
CSFV Brazil 2019-571 (Genotype 1.5)	12	−1	ND	ND	ND	19.41	18.52	18.22
	13		ND	ND	ND	16.74	14.2	15.52
	16	2	ND	ND	ND	18.05	16.14	16.23
	19		38.11	ND	ND	14.17	14.74	15.46
	24		ND	ND	ND	14.50	14.07	15.26
	4	4	29.46	31.48	32.71	15.05	14.82	15.01
	18		30.49	33.40	34.78	18.60	15.65	16.42
	23		32.26	33.38	34.26	15.01	13.47	14.40
	1	6	31.1	32.00	33.20	18.96	15.59	17.14
	3		26.59	27.73	29.05	17.34	15.21	16.30
	5		26.49	28.00	30.40	18.45	15.05	16.47
	6	8	25.08	26.61	27.93	19.03	19.27	17.11
	7		28.37	29.19	30.51	17.47	14.05	15.23
	11		25.04	25.84	28.05	18.45	15.26	15.56

**Table 6 pathogens-14-00767-t006:** Virus isolation from 2 dpi samples from the CSFV Pinillos and CSFV Brazil 2019-571 experiments. ND = No detection. TCID_50_ = Tissue culture infectious dose 50%.

Strain and Genotype	Pig Number	DPI	CSFV-FLI			Virus Titer Log (TCID_50_/mL)		
			Homo	Fresh Swab	Post-Thaw Swab	Homo	Fresh Swab	Post-Thaw Swab
CSFV Pinillos (Genotype 2.6)	3	2	37.77	ND	ND	2.80	ND	ND
	21		33.16	34.60	33.80	4.10	3.87	3.97
	23		34.15	34.92	36.23	3.97	3.87	3.63
CSFV Brazil (Genotype 1.5)	16	2	ND	ND	ND	ND	ND	ND
	19		38.11	ND	ND	2.63	ND	ND
	24		ND	ND	ND	ND	ND	ND

## Data Availability

The data presented in this study are available upon request from the corresponding author.

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
