# Peer review of "Evaluation of Spleen Swabs for Sensitive and High-Throughput Detection of Classical Swine Fever Virus"

_pathogens, 2025, doi:10.3390/pathogens14080767_

Round 1
Reviewer 1 Report
Comments and Suggestions for Authors
The author Aruna Ambagala and group introduce a high-throughput detection method for classical swine fever (CSF). CSFV genomic material can be detected in spleen swabs instead of spleen homogenates. The viruses can be successfully isolated from the swabs. Especially, the method of preparation swabs from spleen samples were induced in detail and easy to imitate. There are some minor aspects that need further improvement.
1. Is there any difference between RT-PCR and RRT-PCR.
2. All the reagents do not show item numbers.
3. Lines 135-141: Is 45 cycles for RRT-PCR excessive?A Ct value ≥40 is considered negative, and perhaps 35 cycles are sufficient to yield a negative result.
4. The "dpi 2" in line 323 and "TCID50" in line 351 do not follow the standard format.
5. Lines 330-332:The formats of ß-actin and ß-Actin should be consistent.
6. The titile of reference 5,6,7,8.9.10.11 are not consistent to others. And references are too less.
Reviewer 2 Report
Comments and Suggestions for Authors
In this manuscript, Hochman et al, addressed the “Evaluation of Spleen Swabs for Sensitive and High-Throughput Detection of Classical Swine Fever Virus”. Having examined the manuscript, I note that though it discusses interesting observations, to be considered for MDPI Pathogens, the following are some of the comments that the authors might find useful for future submission. This manuscript is well-structured and addresses an important diagnostic challenge in swine health surveillance. It offers practical benefits such as reduced labor and higher throughput. This type of studies are extremely valuable for the scientific community at a global level.
Reviewer Comments
- The authors need to clarify about how successful was the virus isolation from spleen swabs, and how does it compare to isolation from homogenates?
- The authors should clarify the practical applicability of spleen swabs. Can this method be easily implemented in field conditions or routine surveillance programs? A brief discussion on feasibility and limitations would strengthen the manuscript.
- Did you do any statistical comparison between swabs and homogenates? If yes, please include the results.
- What were the conditions (time after infection, storage, etc.) when the swabs gave the best results?
- Were there any differences in detection between fresh and frozen spleen samples?
